# ON THE USE OF WORD EMBEDDINGS ALONE TO REPRESENT NATURAL LANGUAGE SEQUENCES

## ABSTRACT

To construct representations for natural language sequences, information from two main sources needs to be captured: ($i$) semantic meaning of individual words, and ($ii$) their compositionality. These two types of information are usually represented in the form of word embeddings and compositional functions, respectively. For the latter, Recurrent Neural Networks (RNNs) and Convolutional Neural Networks (CNNs) have been considered. There has not been a rigorous evaluation regarding the relative importance of each component to different text-representation-based tasks; *i.e.*, how important is the modeling capacity of word embeddings alone, relative to the added value of a compositional function? In this paper, we conduct an extensive comparative study between Simple Word Embeddings-based Models (SWEMs), with no compositional parameters, relative to employing word embeddings within RNN/CNN-based models. Surprisingly, SWEMs exhibit comparable or even superior performance in the majority of cases considered. Moreover, in a new SWEM setup, we propose to employ a max-pooling operation over the learned word-embedding matrix of a given sentence. This approach is demonstrated to extract complementary features relative to the averaging operation standard to SWEMs, while endowing our model with better interpretability. To further validate our observations, we examine the information utilized by different models to make predictions, revealing interesting properties of word embeddings.

## 1 INTRODUCTION

Word embeddings, learned from massive unstructured text data, are widely-adopted building blocks for Natural Language Processing (NLP). By representing each word as a fixed-length vector, these embeddings can group semantically similar words, while explicitly encoding rich linguistic regularities and patterns (Bengio et al., 2003; Mikolov et al., 2013; Pennington et al., 2014). In the spirit of learning *distributed* representations for natural language, many NLP applications also benefit from encoding word sequences, *e.g.*, a sentence or document, into a fixed-length feature vector. Examples of this are sentence/document classification (Le & Mikolov, 2014; Zhang et al., 2015), text-sequence matching (Hu et al., 2014; Shen et al., 2017), machine translation (Bahdanau et al., 2014), etc. Many architectures have been proposed to model the compositionality in variable-length text, leveraging the word-embedding construct. These methods range from simple operations like addition (Mitchell & Lapata, 2010; Iyyer et al., 2015) to more sophisticated compositional functions such as Recurrent Neural Networks (RNNs) (Tai et al., 2015; Sutskever et al., 2014), Convolutional Neural Networks (CNNs) (Kalchbrenner et al., 2014; Kim, 2014) and recursive neural networks (Socher et al., 2011a).

Although those models with more expressive compositional functions, *e.g.*, recurrent or convolutional networks, have demonstrated impressive results, they are typically computationally expensive, due to the need to estimate hundreds of thousands, if not millions, of parameters (Parikh et al., 2016). In constrast, models with simple compositional functions often compute a sentence or document embedding by simply taking the summation, or averaging, over the word embedding of each sequence element obtained via, *e.g.*, *word2vec* (Mikolov et al., 2013), or *GloVe* (Pennington et al., 2014). Generally, such a Simple Word Embedding-based Model (SWEM) does not explicitly account for word-order information within a text sequence. However, they possess the desirable property of having significantly fewer parameters and much faster training, relative to recurrent- or convolutional-based models. Hence, there is a computation-*vs.*-expressiveness tradeoff regarding

how to model the compositionality of a text sequence. Moreover, it is of interest to examine the practical (empirical) value of the additional expressiveness, on many standard NLP problems.

Recently, several studies suggest that on certain NLP applications much simpler word embedding-based architectures exhibit comparable or even superior performance, compared with more complicated models using recurrence or convolutions. For instance, Parikh et al. (2016) employed a decomposable attention mechanism operating on the word embedding layer, achieving state-of-the-art results on the Stanford Natural Language Inference (SNLI) corpus (Bowman et al., 2015), with considerably fewer parameters. More recently, Vaswani et al. (2017) developed a network architecture for machine translation solely based on attention, without recurrence or convolutions, that yielded state-of-the-art BLEU scores on the English-to-German translation task. Although complex compositional functions are avoided in these models, additional modules, such as attention layers, are employed on top of the word embedding layer. As a result, the specific role that the word embeddings plays in these models is not emphasized (or explicit), which distracts from understanding how important the word embeddings alone are to the observed superior performance.

More importantly, from the perspective of representing natural language sequences, existing work (Wieting et al., 2015; Arora et al., 2016; Parikh et al., 2016) only compared simple compositional functions with an LSTM (Long Short-Term memory) or CNN on a limited set of tasks, while mostly focusing on fairly short sentences (up to approximately 50 words). However, as indicated in Wieting et al. (2015), the superiority of recurrent or convolutional compositional architectures is highly dependent on the nature of specific applications, such as text length, task goal, etc.

**Our Contribution** In this paper, we conduct an extensive experimental investigation regarding the ability of word embeddings to represent sentences or (longer) documents. The principal motivation is to understand whether word embeddings themselves already carry sufficient information for the corresponding prediction on a variety of NLP tasks. To emphasize the expressiveness of word embeddings, we compare several simple word embeddings-based models, which have no compositional parameters, with existing recurrent and convolutional networks, in a point-by-point manner. Specifically, we consider three tasks with distinct properties: *document classification* (Yahoo news, Yelp reviews, etc.), *(short) sentence classification* (Stanford sentiment treebank, TREC, etc.), and *natural language sequence matching* (SNLI, WikiQA, etc.). Moreover, we propose to leverage a new *max-pooling* operation over the word embedding representation of given text, which is demonstrated in our experiments to extract complementary features relative to the averaging operation. As a side benefit, the max-pooling operation also endows our model with better interpretability. Meaningful semantic structures are manifested in the learned word embeddings, that shed light on the prediction process of our models.

To gain better insight into the properties of word embeddings, and SWEM architectures, we further explore the sensitivity of different compositional functions to the size of the training data, by comparing SWEM with CNN and LSTM in cases where only a subset of the original training set samples are available. In order to validate our experimental findings, we conduct additional experiments to understand how much of the word-order information is utilized to make the corresponding prediction on different tasks. We also investigate the dimensionality of word embeddings required for SWEM to be sufficiently expressive.

**Limitations** Our investigation regarding the modeling capacity of word embeddings also has limitations. First, we examine the most basic, yet representative, forms of one-layer recurrent/convolutional models for comparisons and do not consider other sophisticated model variants. Thus, our conclusions are limited to algorithms explored in this paper. Where available from the literature, we *do* compare to some deep models, such as the deep CNN construct. Additional modules (such as attention layers) can also be combined with our SWEM to yield better performance, which is not the main goal of this study (as we wish to focus on the word embeddings themselves), and is thus left for future work. Second, our discussion only considers NLP problems defined by the datasets considered, which may not fully capture the difficulty of representing and reasoning over natural language sequences. However, our exploration covers a wide variety of real-world applications (with large-scale datasets) and thus, we hypothesize our conclusions should be representative of the English language in many cases of interest.

**Summary of Findings** Keeping these limitations in mind, our findings regarding when (and why) word embeddings are enough for text sequence representations are summarized as follows:

- Word embeddings are surprisingly effective at representing longer documents (with hundreds of words), while recurrent/convolutional compositional functions are necessary when constructing representations for short sentences.

- The SWEM architecture performs stronger on topic categorization tasks than on sentiment analysis, due to the different levels of sensitivity to word-order information for the two tasks.

- To match natural language sentences, *e.g.*, textual entailment, answer sentence selection, etc., word embeddings are already sufficiently informative for the corresponding prediction, while adopting complicated compositional functions like LSTM or CNN tends to be substantially less helpful.

- For our SWEM-*max* model (employing a max pooling within SWEM), each dimension of the word embedding contains interpretable semantic patterns, and groups together words with a common theme or *topic*.

- SWEMs are much less likely to overfit than an LSTM or CNN, with training data of limited size, exhibiting superior performance even with only *hundreds* of training observations.

- SWEMs demonstrate competitive results with small word-embedding dimensions, suggesting that word embeddings are *efficient* at encoding semantic information.

## 2   RELATED WORK

A fundamental goal in NLP is to develop expressive, yet computationally efficient compositional functions that can capture the linguistic structure of natural language sequences. A variety of models have been proposed to account for different properties of text sequences, which may be divided into two main categories: (*i*) *simple compositional functions* that largely leverage information from the word embeddings to extract semantic features; and (*ii*) *complex compositional functions* that construct words into text representations in a recurrent or convolutional manner, and can in principle capture the word-order features either globally or locally. However, several recent studies have shown empirically that the advantages of distinct compositional functions are highly dependent on the specific task (Mitchell & Lapata, 2010; Iyyer et al., 2015; Wieting et al., 2015; Arora et al., 2016; Vaswani et al., 2017; Parikh et al., 2016). This is intuitively reasonable since different properties of a text sequence may be required, depending on the nature of specific problems. However, previous research only focused on one or two problems at a time, thus a comprehensive study regarding the effectiveness of various compositional functions on distinct NLP tasks, *e.g.*, categorizing short sentence/long documents, matching natural language sentences, has heretofore been absent. Our work seeks to perform a comprehensive comparison with respect to these two types of compositional functions, across a wide range of NLP problems, and reveals some general rules for rationally selecting models to tackle different tasks.

## 3   MODELS & TRAINING

Consider a text sequence $\boldsymbol{X}$ (either a sentence or a document), composed of a sequence of words: $\{w_1, w_2, ...., w_L\}$, where $L$ is the number of tokens, *i.e.*, the sentence/document length. Let $\{v_1, v_2, ...., v_L\}$ denote the respective word embedding for each token, where $v_l \in \mathbb{R}^K$ and $K$ is the dimensionality of the embedding. The compositional function, $\boldsymbol{X} \rightarrow \boldsymbol{z}$, aims to combine the word embeddings into a fixed-length sentence/document representation $\boldsymbol{z}$. In the following, we describe different types of functions to be considered in this work.

### 3.1   SIMPLE WORD-EMBEDDING BASED MODEL (SWEM)

To investigate the modeling capacity of word embeddings, we consider a type of model with no additional compositional parameters to encode natural language sequences, termed a SWEM. Among them, the simplest strategy is to compute the element-wise average over word vectors for a given sequence (Wieting et al. (2015); Adi et al. (2016)):

$$\boldsymbol{z} = \frac{1}{L} \sum_{i=1}^{L} v_i \, . \tag{1}$$

The model in (1) averages over each of the $K$ dimensions for all words, resulting in a representation $z$ with the same dimension as word embeddings (termed SWEM-*aver*). Intuitively, $z$ takes the information of every sequence element into account using the addition operation.

Motivated by the success of employing pooling layers to down-sample representations for image data (Krizhevsky et al. (2012)), we propose another SWEM variant, that extracts the most salient features from every word embedding dimension, by taking the maximum value along each dimension of the word vectors. This strategy is also similar to the max-over-time pooling operation in convolutional neural networks (Collobert et al., 2011):

$$z = \text{max-pooling}(v_1, v_2, ..., v_L) \,. \tag{2}$$

We denote this model variant as SWEM-*max*. Here the $j$-th component of $z$ is the maximum element in the set $\{v_{1j}, \ldots, v_{Lj}\}$, where $v_{1j}$ is, for example, the $j$-th component of $v_1$. Considering that SWEM-*aver* and SWEM-*max* are complementary, in the sense that they account for different types of information from text sequences, we also propose a third SWEM variant, where the two abstracted features are concatenated together to form the sentence embeddings (denoted as SWEM-*concat*). It is worth noting that for all SWEM variants, there are no additional compositional parameters to be learned. As a result, models can only exploit intrinsic word embedding information for predictions.

## 3.2 Recurrent Sequence Encoder

A widely adopted compositional function is defined in a recurrent manner: the model successively takes word vector $v_t$ at step $t$, along with hidden unit $h_{t-1}$ from the last time step, to update the hidden state via $h_t = f(v_t, h_{t-1})$, where $f(\cdot)$ is the transition function. To address the issue of learning long-term dependencies, $f(\cdot)$ is often defined as Long Short-Term Memory (LSTM) (Hochreiter & Schmidhuber, 1997), which employs *gates* ($o_t$, $f_t$ and $i_t$, as output, forget and input gates, respectively) to control the information abstracted from a sequence using:

$$\begin{bmatrix} i_t \\ f_t \\ o_t \\ \tilde{c}_t \end{bmatrix} = \begin{bmatrix} \sigma \\ \sigma \\ \sigma \\ \tanh \end{bmatrix} \left( \boldsymbol{W} \cdot \begin{bmatrix} h_{t-1} \\ v_t \end{bmatrix} \right), \qquad c_t = f_t \odot c_{t-1} + i_t \odot \tilde{c}_t, \qquad h_t = o_t \odot c_t,$$

where $\odot$ stands for element-wise (Hadamard) multiplication. The last hidden state $h_L$ or the average over all hidden states, $h_1, \ldots, h_L$, is typically utilized as the final representation $z$. Intuitively, the LSTM encodes a text sequence considering its word-order information, but yields additional compositional parameters, $\boldsymbol{W}$, that must be learned.

## 3.3 Convolutional Sequence Encoder

The Convolutional Neural Network (CNN) architecture in Kim 2014; Collobert et al. 2011; Gan et al. 2017 is another strategy extensively employed as the compositional function for encoding text sequences. The convolution operation considers every window of $n$ words within the sequence $\boldsymbol{X}$, i.e., $\{w_{1:n}, w_{2:n+1}, ..., w_{L-n+1:L}\}$. These $n$-gram text subsequences can be represented by the concatenation of all corresponding word vectors, i.e., $\{v_{1:n}, v_{2:n+1}, ..., v_{L-n+1:L}\}$. A filter $\boldsymbol{U} \in \mathbb{R}^{K \times n}$ is then applied to each word window to generate the corresponding feature:

$$s_i = g(\boldsymbol{U} \odot v_{i:i+n-1} + b),$$

where $g(\cdot)$ is a nonlinear function such as hyperbolic tangent and $b$ is a bias term. The features produced by each word window, $s_i$, are concatenated together as a feature map: $s = [s_1, s_2, ..., s_{L-n+1}]$. Subsequently, an aggregation operation such as max-pooling is used on top of the feature maps to abstract the most salient semantic features, resulting in the final representation $z$. Multiple learned filters are employed, and these may employ different temporal lengths $n$. For simplicity we have discussed a single-layer CNN text model. Deep CNN text models have also been developed (Conneau et al., 2016), and we perform empirical comparisons to such models below.

## 3.4 Parameters & Computation Comparison

We compare CNN, LSTM and SWEM w.r.t. their parameters and computational speed. $K$ denotes the dimension of word embeddings, as above. For the CNN, we use $n$ to denote the filter width (assumed the same for all filters, for simplicity of analy-

| Model | Parameters | Complexity | Seq. Operations |
|-------|------------|------------|-----------------|
| CNN | $n \cdot K \cdot d$ | $\mathcal{O}(n \cdot L \cdot K \cdot d)$ | $\mathcal{O}(1)$ |
| LSTM | $4 \cdot d \cdot (K + d)$ | $\mathcal{O}(L \cdot d^2 + L \cdot K \cdot d)$ | $\mathcal{O}(L)$ |
| SWEM | $0$ | $\mathcal{O}(L \cdot K)$ | $\mathcal{O}(1)$ |

Table 1: Comparisons of CNN, LSTM and SWEM architectures. Columns correspond to the number of *compositional* parameters, computational complexity and sequential operations, respectively.

sis, but in practice variable $n$ may be used among the CNN filters). We define $d$ as the dimension of the final sequence representation. Specifically, $d$ represents the dimension of hidden units or the number of filters in LSTM or CNN, respectively. We first examine the number of *compositional parameters* for each model. As shown in Table 1, both the CNN and LSTM have a large number of parameters, to model the semantic compositionality of text sequences, whereas SWEM has no such parameters. Similar to Vaswani et al. (2017), we then consider the computational complexity and the minimum number of sequential operations required for each model. SWEM tends to be more efficient than CNN and LSTM in terms of computation complexity. For example, considering the case where $K = d$, SWEM is faster than CNN or LSTM by a factor of $nd$ or $d$, respectively. Further, the computations in SWEM are highly parallelizable, unlike LSTM that requires $\mathcal{O}(L)$ sequential steps.

## 4 EXPERIMENTS

We evaluate different compositional functions on a wide variety of supervised tasks, including document categorization, text sequence matching (given a sentence pair, $\boldsymbol{X_1}$, $\boldsymbol{X_2}$, predict their relationship, $y$) as well as (short) sentence classification. We experiment on 15 datasets regarding natural language understanding, with corresponding data statistics summarized in the Supplementary Material. Our code will be released to encourage future research.

We use 300-dimensional GloVe word embeddings (Pennington et al., 2014) as initialization for all our models. Out-Of-Vocabulary (OOV) words are initialized from a uniform distribution with range $[-0.01, 0.01]$. The GloVe embeddings are employed in two ways for learning the refined word embeddings: ($i$) directly updating each word embedding during training; and ($ii$) training a 300-dimensional multilayer perceptron (MLP) layer with ReLU activation, with GloVe embeddings input to the MLP and with output defining the updated word embeddings. This latter approach corresponds to learning an MLP model that transforms GloVe embeddings to the dataset and task of interest. The advantages of these two methods differs from dataset to dataset. We choose the better strategy based on their corresponding performances on the validation set. The final classifier is implemented as an MLP layer with dimension selected from the set $[100, 300, 500, 1000]$, followed by a sigmoid or softmax function depending on the specific task.

Adam (Kingma & Ba, 2014) is used to optimize all models, with learning rate selected from the set $[1e-3, 3e-4, 2e-4, 1e-5]$ (with cross-validation used to select the appropriate parameter for a given dataset and task). Dropout regularization (Srivastava et al., 2014) is employed on the word embedding layer and final MLP layer, with the dropout rate selected from the set $[0.2, 0.5, 0.7]$. The batch size is selected from $[2, 8, 32, 128, 512]$.

### 4.1 DOCUMENT CATEGORIZATION

We begin with the task of categorizing documents (with approximately 100 words in average per document). We follow the data split in Zhang et al. (2015) for comparability. These datasets can be generally categorized into three types: *topic categorization* (represented by Yahoo! Answer and AG news), *sentiment analysis* (represented by Yelp Polarity and Yelp Full) and *ontology classification* (represented by DBpedia). Results are shown in Table 2. Surprisingly, on topic prediction tasks, our SWEM model exhibits stronger performances, relative to both LSTM and CNN compositional architectures, this by leveraging both the average and max-pooling features from word embeddings. Specifically, our SWEM-*concat* model even outperforms a 29-layer deep CNN model (Conneau et al., 2016) when predicting topics. On the ontology classification problem (DBpedia dataset), we

| Model | Yahoo! Ans. | AG News | Yelp P. | Yelp F. | DBpedia |
|---|---|---|---|---|---|
| Bag-of-means[*] | 60.55 | 83.09 | 87.33 | 53.54 | 90.45 |
| Small word CNN[*] | 69.98 | 89.13 | 94.46 | 58.59 | 98.15 |
| Large word CNN[*] | 70.94 | 91.45 | 95.11 | 59.48 | 98.28 |
| LSTM[*] | 70.84 | 86.06 | 94.74 | 58.17 | 98.55 |
| Deep CNN (29 layer)[‡] | 73.43 | 91.27 | **95.72** | **64.26** | **98.71** |
| SWEM-*aver* | 73.14 | 91.71 | 93.59 | 60.66 | 98.42 |
| SWEM-*max* | 72.66 | 91.79 | 93.25 | 59.63 | 98.24 |
| SWEM-*concat* | **73.53** | **92.24** | 93.76 | 61.11 | 98.57 |

Table 2: Test error rates on (long) document classification tasks, in percentage. Results marked with ∗ are reported in Zhang et al. (2015), with † are reported in Dai & Le (2015), and with ‡ are reported in Conneau et al. (2016).

observe the same trend, that SWEM exhibits comparable or even superior results, compared with CNN or LSTM models.

Since there are no compositional parameters in SWEM, our models have an order of magnitude fewer parameters (excluding embeddings) than LSTM or CNN, and are considerably more computationally efficient. As illustrated in Table 4, SWEM-*concat* achieves better results on Yahoo! Answer than CNN/LSTM, with only 61K parameters (one-tenth the number of LSTM parameters, or one-third the number of CNN parameters), while taking a fraction of the training time relative to the CNN or LSTM.

| Politics | Science | Computer | Sports | Chemistry | Finance | Geoscience |
|---|---|---|---|---|---|---|
| philipdru | coulomb | system32 | billups | sio2 ($SiO_2$) | proprietorship | fossil |
| justices | differentiable | cobol | midfield | nonmetal | ameritrade | zoos |
| impeached | paranormal | agp | sportblogs | pka | retailing | farming |
| impeachment | converge | dhcp | mickelson | chemistry | mlm | volcanic |
| neocons | antimatter | win98 | juventus | quarks | budgeting | ecosystem |

Table 3: Top five words with the largest values w.r.t. a give word embeddings' dimension (each column corresponds to a dimension). The first row shows the topic for words in each column.

| Model | Parameters | Speed |
|---|---|---|
| CNN | 541K | 171s |
| LSTM | 1.8M | 598s |
| SWEM | **61K** | **63s** |

Table 4: Speed & Parameters on Yahoo! Answer dataset.

However, for the sentiment analysis tasks, both CNN and LSTM compositional functions perform better than SWEM, suggesting that word-order information may be required for analyzing sentiment orientations. This finding is consistent with Pang et al. (2002), where they hypothesize that the positional information of a word in text sequences may be beneficial to predict sentiment. This is intuitively reasonable since, for instance, the phrase 'not really good' and 'really not good' convey different levels of negative sentiment, while being different only by their word orderings. Contrary to SWEM, CNN and LSTM models can both capture this type of information via convolutional filters or recurrent transition functions. However, as suggested above, such word-order patterns may be much less useful for predicting the topic of a document. This may be attributed to the fact that word embeddings alone already provide sufficient topic information of a document, at least when the text sequences considered are relatively long.

Although the proposed SWEM-*max* variant generally performs a bit worse than SWEM-*aver*, it extracts complementary features from SWEM-*aver*, and hence in most cases SWEM-*concat* exhibits the best performance among all SWEM variants. Further, we found that the word embeddings learned from SWEM-*max* tend to be very sparse. We trained our SWEM-*max* model on the Yahoo datasets (randomly initialized from a uniform distribution with range $[0, 0.001]$). With the learned embeddings, we plot the values for each of the word embedding dimensions, for the entire vocabulary. As shown in Figure 1, most of the embedding values are highly concentrated around zero, indicating that the word embeddings learned are very sparse. By contrast, the GloVe word embeddings, for the same vocabulary, are much more dense than the embeddings learned from SWEM-*max*. This suggests that the model may only depend on a few key words, among the entire vocabulary, for predictions (since most words do not contribute to the summation or max operation in SWEM). Through the embedding, the model learns the important words for a given task (those words with non-zero embedding components).

| Model | SNLI | MultiNLI | | WikiQA | | Quora | MSRP | |
|---|---|---|---|---|---|---|---|---|
| | | Matched | Mismatched | | | | | |
| | *Acc.* | *Acc.* | *Acc.* | *MAP* | *MRR* | *Acc.* | *Acc.* | *F1* |
| CNN | 82.1 | 65.0 | 65.3 | 0.6752 | 0.6890 | 79.60 | 69.9 | 80.9 |
| LSTM | 80.6 | 66.9* | 66.9* | **0.6820** | **0.6988** | 82.58 | 70.6 | 80.5 |
| SWEM-*aver* | 82.3 | 66.5 | 66.2 | **0.6808** | **0.6922** | 82.68 | 71.0 | 81.1 |
| SWEM-*max* | **83.8** | **68.2** | **67.7** | 0.6613 | 0.6717 | 82.20 | 70.6 | 80.8 |
| SWEM-*concat* | 83.3 | 67.9 | 67.6 | 0.6788 | 0.6908 | **83.03** | **71.5** | **81.3** |

Table 5: Performance of different models on matching natural language sentences. Results with * are for Bidirectional LSTM, reported in Williams et al. (2017). Our reported results on MultiNLI are only trained MultiNLI training set (without training data from SNLI). For MSRP dataset, we follow the setups in Hu et al. (2014) and do not use any additional (hand-crafted) features.

Moreover, the nature of the max-pooling process gives rise to a more interpretable model. For a document, only the word with largest value in each embedding dimension is employed for the final representation. In this regard, we suspect that semantically similar words may have large values in some shared dimensions. So motivated, after training the SWEM-*max* model on the Yahoo dataset, we selected five words with the largest values, among the entire vocabulary, for each word embedding dimension (these words are selected preferentially in the corresponding dimension, by the max operation). As shown in Table 3, the words chosen w.r.t. each embedding dimension are indeed highly relevant and correspond to a common topic (the topics are inferred from words). For example, the words in the first column of Table 3 are all political terms, which could be assigned to the *Politics & Government* topic. Note that our model can even learn locally interpretable structure that is not explicitly indicated by the label information. For instance, all words in the fifth column are *Chemistry*-related. However, we do not have a chemistry label in the dataset, and regardless they should belong to the *Science* topic.

Moreover, we summed up all embedding dimensions for each word in the vocabulary and selected 20 words with the largest total value. We assume that these words should be highly predictive since they are more likely to "survive" the max-pooling operation. These words are listed as below:

*'askcomputerexpert', 'midfield', 'presario', 'preventdisease', 'dhcp', 'playgolfamerica', 'radeon', 'win32', 'system32', 'colston', 'juventus', 'mayweather', 'murtha', 'hoodia', 'lebron', 'theist', 'billups', cannavaro', 'maldini', 'ronaldhino'.*

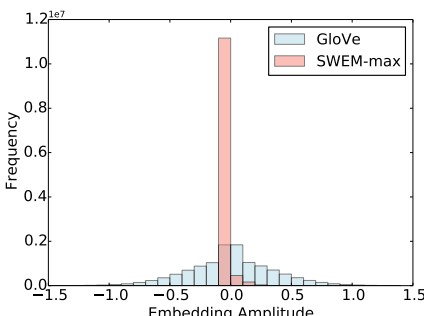

Figure 1: The histograms for learned word embeddings (randomly initialized) of SWEM-*max* and GloVe embeddings for the same vocabulary, trained on Yahoo! Answer dataset.

These words can be generally grouped into two categories: the first are the names of sports players/teams (*e.g.*, 'ronaldhino', 'lebron' or 'juventus'), software product/brand (*e.g.*, 'win32', 'radeon') or plants (*e.g.* 'hoodia'). These words are important since their occurence may already indicate the assigned label. The second are field-specific terms regarding a topic, such as 'askcomputerexpert' to the *Computers & Internet* topic, 'preventdisease' to the *Health* topic or 'midfield' to the *Sports* topic. Again, these words are likely to occur in documents with matching topic.

## 4.2 TEXT SEQUENCE MATCHING

To gain a deeper understanding regarding the modeling capacity of word embeddings, we further investigate the problem of sentence matching, including natural language inference, answer sentence selection and paraphrase identification. The corresponding performances are shown in Table 5. Surprisingly, on most of the datasets considered (except WikiQA), SWEM demonstrates the best results compared with those with CNN or the LSTM encoder. Notably, on SNLI dataset, we observe that SWEM-*max* performs the best among all SWEM variants, consistent with the findings in Nie & Bansal (2017); Conneau et al. (2017) that *max-pooling* over BiLSTM hidden units outperforms average pooling operation on SNLI dataset. As a result, with only 120K parameters, our SWEM-*max* achieves a test accuracy of 83.8%, which is very competitive among state-of-the-art sentence encoding-based models (in terms of both performance and number of parameter)[1].

---

[1]See leaderboard at `https://nlp.stanford.edu/projects/snli/` for details.

The strong results of the SWEM setup on these tasks may stem from the fact that when matching natural language sentences, it is sufficient in most cases to simply model the word-level alignments between two sequences (Parikh et al., 2016). From this perspective, word-order information becomes much less useful for predicting relationship between sentences. Moreover, considering the simpler model architecture of SWEM, they could be much easier to be optimized than LSTM or CNN-based models, and thus give rise to better empirical results.

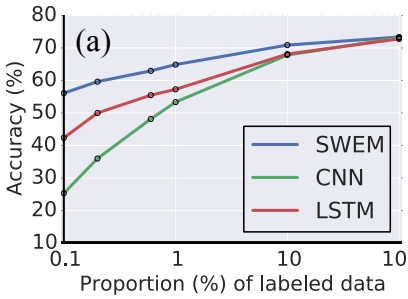 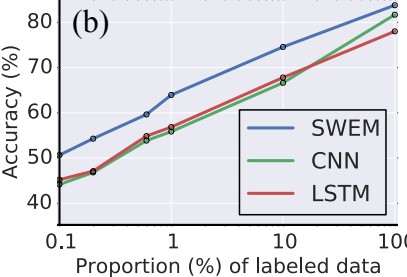

Figure 2: The test accuracy comparisons between SWEM and CNN/LSTM on (a) Yahoo! Answers dataset and (b) SNLI dataset, with different proportions of training data (ranging from 0.1% to 100%).

To explore the robustness of different compositional functions, we consider another application scenario, where we only have a limited number of training data, *e.g.*, when labeled data are expensive to obtain. To investigate this, we re-run the experiments on Yahoo and SNLI datasets, while employing increasing proportions of the original training set. Specifically, we use 0.1%, 0.2%, 0.6%, 1.0%, 10%, 100% for comparison; the corresponding results are shown in Figure 2. Surprisingly, SWEM consistently outperforms CNN and LSTM models by a large margin, on a wide range of training data proportions. For instance, with 0.1% of the training samples from Yahoo dataset (around 1.4K labeled data), SWEM achieves an accuracy of 56.10%, which is much better than that of models with CNN (25.32%) or LSTM (42.37%). On the SNLI dataset, we also noticed the same trend that the SWEM architecture result in much better accuracies, with a fraction of training data. This observation indicates that overfitting issues in CNN or LSTM-based models on text data mainly stems from over-complicated compositional functions, rather than the word embedding layer. More importantly, SWEM tends to be a far more robust model when only limited data are available for training.

## 4.3 SHORT SENTENCE CLASSIFICATION

We now consider sentence-classification tasks (with approximately 20 words on average). We experiment on three sentiment classification datasets, *i.e.*, MR, SST-1, SST-2, as well as subjectivity classification (Subj) and question classification (TREC). The corresponding results are shown in Table 6. Compared with CNN/LSTM compositional functions, SWEM yields inferior accuracies on sentiment analysis datasets, consistent with our observation in the case of document categorization. However, SWEM exhibits comparable performance on other two tasks, again with much less parameters and faster training. Generally, SWEM is less effective at extracting representations from (short) sentences than from (long) documents. This may be due to the fact that for a shorter text sequence, word-order features tend to be more important since the semantic information provided by word embeddings is relatively limited.

| Model | MR | SST-1 | SST-2 | Subj | TREC |
|---|---|---|---|---|---|
| RAE (Socher et al. (2011b)) | 77.7 | 43.2 | 82.4 | – | – |
| MV-RNN (Socher et al. (2012)) | 79.0 | 44.4 | 82.9 | – | – |
| LSTM (Tai et al. (2015)) | – | 46.4 | 84.9 | – | – |
| RNN (Zhao et al. (2015)) | 77.2 | – | – | **93.7** | 90.2 |
| Dynamic CNN (Kalchbrenner et al. (2014)) | – | 48.5 | 86.8 | – | 93.0 |
| CNN (Kim (2014)) | **81.5** | **48.0** | **88.1** | 93.4 | **93.6** |
| SWEM-*aver* | 77.6 | 45.2 | 83.9 | 92.5 | **92.2** |
| SWEM-*max* | 76.9 | 44.1 | 83.6 | 91.2 | 89.0 |
| SWEM-*concat* | **78.2** | **46.1** | **84.3** | **93.0** | 91.8 |

Table 6: Test accuracies with different compositional functions on (short) sentence classifications.

Moreover, we note that the results on these relatively small datasets are highly sensitive to model regularization techniques due to the overfitting issues. In this regard, one interesting future direction may be to develop specific regularization strategies for the SWEM setup, and thus make them work better on small sentence classification datasets.

| Negative: | Friendly staff and nice selection of vegetarian options . Food is just okay , not great. Makes me wonder why everyone likes food fight so much. |
|---|---|
| Positive: | The store is small , but it carries specialties that are difficult to find in Pittsburgh. I was particularly excited to find middle eastern chili sauce and chocolate covered turkish delights. |
| Negative: | If you love long lines and only 4 or less lanes open, then this is the place to be. The lines are long and the cashiers are usually old people who take their time with everything. |

Table 8: The test samples from Yelp Polarity dataset that LSTM gives wrong predictions with shuffled training data, but predicts correctly with the original training set. Therefore, word order should be relatively important in these cases for predicting the corresponding sentiment (the first column shows the ground truth labels).

## 5 PROPERTIES OF WORD EMBEDDINGS

To further reveal the modeling capacity of word embeddings to represent natural language sequences, we perform additional experiments to answer the following interesting questions:

| Datasets | Yahoo | Yelp P. | SNLI |
|---|---|---|---|
| Original | 72.78 | 95.11 | 78.02 |
| Shuffled | 72.89 | 93.49 | 77.68 |

Table 7: Test accuracy for LSTM model trained on original/shuffled training set.

**How important is word-order information for distinct tasks?** One possible disadvantage of SWEM is that it ignores the word-order information within a text sequence, which could be potentially captured by CNN- or LSTM-based models. However, we empirically found that except for sentiment analysis, SWEM exhibits similar or even superior performances than CNN or LSTM on a variety of tasks. In this regard, one natural question would be: how important are word-order features for these tasks?

To this end, we randomly shuffle the words for every sentence in the training set, while keeping the original word order for samples in the test set. The motivation here is to remove the word-order features from the training set and examine how sensitive the performance on different tasks are to word-oder information. We use LSTM as the model for this purpose since it can captures word-order information from the original training set. The results on three distinct tasks are shown in Table 7. Somewhat surprisingly, for Yahoo and SNLI datasets, the LSTM model trained on shuffled training set shows comparable accuracies to those trained on the original dataset, indicating that word-order information does not contribute significantly on these two problems, *i.e.*, topic categorization and textual entailment. However, on the Yelp polarity dataset, the results drop noticeably, further suggesting that word-order does matter for sentiment analysis (as indicated above from a different perspective).

Notably, the performance of LSTM on the Yelp dataset with a shuffled training set is very close to our results with SWEM, indicating that the main difference between LSTM and SWEM may be due to the ability of the former to capture word-order features. Both observations are in consistent with our experimental results in the previous section.

To understand what type of sentences are sensitive to word-order information, we further show those samples that are mis-predicted because of the shuffling of training data in Table 8. Taking the first sentence as an example, several words in the review are generally positive, *i.e. friendly*, *nice*, *okay*, *great* and *likes*. However, the most vital features for predicting the sentiment of this sentence could be the phrase/sentence *'is just okay'*, *'not great'* or *'makes me wonder why everyone likes'*, which cannot be captured without considering word-order features.

**How many word embedding dimensions are needed?** Since there are no compositional parameters in SWEM, the component that contains the semantic information of a text sequence is the word embedding. Thus, it

| Embedding dim. | 3 | 10 | 30 | 100 | 300 | 1000 |
|---|---|---|---|---|---|---|
| Yahoo | 64.05 | **72.62** | 73.13 | 73.12 | 73.24 | 73.31 |

Table 9: Test accuracy of SWEM on Yahoo dataset with a wide range of word embedding dimensions.

is of interest to see how many word embedding dimensions are needed for a SWEM architecture to perform well. To this end, we vary the dimension from 3 to 1000 and train a SWEM-*concat* model on the Yahoo dataset. For fair comparison, the word embeddings are randomly initialized in this experiment, since there are no pre-trained word vectors, such as GloVe (Pennington et al., 2014), for some dimensions we consider. As shown in Table 9, the model exhibits higher accuracy with larger word embedding dimensions. This is not surprising since with more embedding dimensions, more semantic features could be potentially encapsulated. However, we also observe that even with only 10 dimensions, SWEM demonstrates comparable results relative to the case with 1000 dimensions, suggesting that word embeddings are very efficient at abstracting semantic information

into fixed-length vectors. This property indicates that we may further reduce the number of model parameters with lower-dimensional word embeddings, while still achieving competitive results.

## 6 CONCLUSION & FUTURE DIRECTIONS

We have performed a comparative study between SWEM (with parameter-free compositional functions) and CNN or LSTM-based models, to represent text sequences on a wide range of NLP tasks. We further validated our experimental findings through additional exploration, and revealed some interesting properties of word embeddings.

Our study regarding the capacity of word embeddings has several implications for future research:

(*i*) The SWEM architecture is a simple, yet very effective strategy to encode text sequences for a wide variety of tasks. We suggest that SWEM should be considered as a strong baseline model while developing other (more sophisticated) neural network architectures.

(*ii*) Additional modules, such as use of an attention mechanism or memory network, could be directly combined with word embeddings to further enhance the model expressiveness, yet preserve the low computational cost (one work along this line could be Parikh et al. (2016)).

(*iii*) Simple manipulation of word embeddings provides new opportunities towards visualizing and rationalizing predictions made by deep learning models.

An important aspect of the SWEM-learned embeddings is that they are very sparse, much more so than the relatively dense embeddings manifested by methods like GloVe. This indicates that only a small fraction of learned key words contribute to the summation and max operations in SWEM-*aver* and SWEM-*max*, respectively. These non-zero components also yield interpretable topics that drive model performance. We observed that the CNN- and LSTM-refined word embeddings are also very sparse. This is an insight that has not been widely noted in the literature, and it may suggest an avenue for interpreting and understanding the success of these classes of NLP methods.

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

## APPENDIX I: EXPERIMENTAL SETUP

### 6.1 DATA STATISTICS

We consider a wide range of text-representation-based tasks in this paper, including *document categorization*, *text sequence matching* and *(short) sentence classification*. The statistics and corresponding types of these datasets are summarized in Table 10

| Datasets | #w | #c | Train | Types |
|---|---|---|---|---|
| Yahoo | 104 | 10 | 1,400K | Topic categorization |
| AG News | 43 | 4 | 120K | Topic categorization |
| Yelp P. | 138 | 2 | 560K | Sentiment analysis |
| Yelp F. | 152 | 5 | 650K | Sentiment analysis |
| DBpedia | 57 | 14 | 560K | Ontology classification |
| SNLI | 11 / 6 | 3 | 549K | Textual Entailment |
| MultiNLI | 21/11 | 3 | 393K | Textual Entailment |
| WikiQA | 7 / 26 | 2 | 20K | Question answering |
| Quora | 13 / 13 | 2 | 384K | Paraphrase identification |
| MSRP | 23 / 23 | 2 | 4K | Paraphrase identification |
| MR | 20 | 2 | 11K | Sentiment analysis |
| SST-1 | 18 | 5 | 12K | Sentiment analysis |
| SST-2 | 19 | 2 | 10K | Sentiment analysis |
| Subj | 23 | 2 | 10K | Subjectivity classification |
| TREC | 10 | 6 | 6K | Question classification |

Table 10: Data Statistics. Where **#w**, **#c** and Train denote the average number of words, the number of classes and the size of training set, respectively. For sentence matching datasets, **#w** stands for the average length for the two corresponding sentences.

### 6.2 WHAT ARE THE KEY WORDS USED FOR PREDICTIONS?

Given the sparsity of word embeddings, one natural question would be: What are those *key words* that are leveraged by the model to make predictions? To this end, after training SWEM-*max* on Yahoo! Answer dataset, we selected the top-10 words (with the maximum values in that dimension) for every word embedding dimension. The results are visualized in Figure 3. These words are indeed very predictive since they are likely to occur in documents with a specific topic, as discussed above. Another interesting observation is that the frequencies of these words are actually quite low in the training set (*e.g. colston*: 320, *repubs*: 255 *win32*: 276), considering the large size of the training set (1,400K). This suggests that the model is utilizing those relatively rare, yet representative words of each topic for the final predictions.

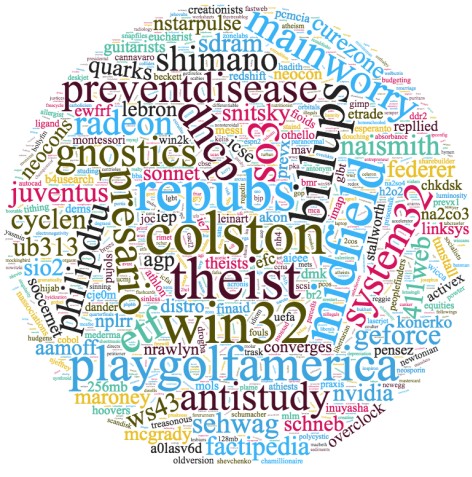

Figure 3: The top 10 words for each word embeddings' dimension.

