# OpenReview forum: "On the Use of Word Embeddings Alone to Represent Natural Language Sequences"
_ICLR.cc/2018/Conference — Reject_

### Official Review · AnonReviewer1 · 2017-11-25
**Review of "On the Use of Word Embeddings..."**

**Rating:** 5
**Confidence:** 4

**Review:**

This paper empirically investigates the differences realized by using compositional functions over word embeddings as compared to directly operating the word embeddings. That is, the authors seek to explore the advantages afforded by RNN/CNN based models that induce intermediate semantic representations of texts, as opposed to simpler (parameter-free) approaches to composing these, like addition.

In sum, I think this is exploration is interesting, and suggests that we should perhaps experiment more regularly with simple aggregation methods like SWEM. On the other hand, the differences across the models is relatively modest, and the data resists clear conclusions, so I'm not sure that the work will be very impactful. In my view, then, this work does constitute a contribution, albeit a modest one. I do think the general notion of attempting to simplify models until performance begins to degrade is a fruitful path to explore, as models continue to increase in complexity despite compelling evidence that this is always needed.

Strengths
---
+ This paper does highlight a gap in existing work, as far as I am aware: namely, I am not sure that there are generally known trade-offs associated with different compositional models over token embeddings for NLP. However, it is not clear that we should expect there to be a consistent result to this question across all NLP tasks.

+ The results are marginally surprising, insofar as I would have expected the CNN/RNN (particularly the former) to dominate the simpler aggregation approaches, and this does not seem borne out by the data. Although this trend is seemingly reversed on the short text data, muddying the story.

Weaknesses
---
- There are a number of important limitations here, many of which the authors themselves note, which mitigate the implications of the reported results. First, this is a small set of tasks, and results may not hold more generally. It would have been nice to see some work on Seq2Seq tasks, or sequence tagging tasks at least.

- I was surprised to see no mention of the "Fixed-Size Ordinally-Forgetting Encoding Method" (FOFE) proposed by Zhang et al. in 2015, which would seem to be a natural point of comparison here, given that it sits in a sweet spot of being simple and efficient while still expressive enough to preserve word-order information. This actually seems like a pretty glaring omission given that it meets many of the desiderata the authors put forward.

- The interpretability angle discussed seems underdeveloped. I'm not sure that being able to identify individual words (as the authors have listed) meaningfully constitutes "interpretability" -- standard CNNs, e.g., lend themselves to this as well by tracing back through the filter activations.

- Some of the questions addressed seem tangential to the main question of the paper -- e.g., word vector dimensionality seems an orthogonal issue to the composition function, and would influence performance for the more complex architectures as well.

Smaller comments
---
- On page 1, the authors write "By representing each word as a fixed-length vector, these embeddings can group semantically similar words, while explicitly encoding rich linguistic regularities and patterns", but actually I would say that these *implicitly* encode such regularities, rather than explicitly.

- "architecture in Kim 2014; Collobert et al. 2011; Gan et al. 2017" -- citation formatting a bit weird here.


*** Update based on author response ***

I have read the authors response and thank them for the additional details.

Regarding the limited set of problems: of course any given work can only explore so many tasks, but for this to have general implications in NLP I would maintain that a standard (structured) sequence tagging task/dataset should have been considered. This is not about the number of datasets, but rather than diversity of the output spaces therein.

I appreciated the additional details regarding FOFE, which as the authors themselves note in their response is essentially a generalization of SWEM.

Overall, the response has not changed my opinion on this paper: I think this (exploring simple representations and baselines) is an important direction in NLP, but feel that the paper would greatly benefit from additional work.

---

> ### Author Response · Authors · 2017-12-24
> **Authors' response to review 1**
>
> Thanks for your constructive feedback!
>
> - Although our paper has discussed a limited set of problems (which is true for almost any research), we argue that we have explored 15 different NLP datasets (detailed information in Supplementary), which should have covered a wide range of real-world application scenarios. More importantly, our work also sheds lights on how SWEM model works and what types of information are needed for distinct tasks. Therefore, we suppose that our conclusions here should be helpful and general in many cases of interest.
>
> For example, if we are solving a text sequence matching problem where word-order information does not contribute a lot (including textual entailment, paraphrase identification, question answering), according to our research, we would know that employing complicated compositions, such as LSTM or CNN, may not be necessary. In this regard, our work reveals several general rules (along with careful analysis) on rationally selecting model for various NLP tasks, which should be useful for future research.
>
> - “Interpretability” definition: we think that there are some misunderstandings here. The key of our “interpretability” here is that we can endow each dimension of word embeddings, learned by SWEM-max, with a topic-specific meaning. That is, embeddings for individual words with a shared semantic topic typically have their largest values in a shared dimension.
>
> We are aware that word embeddings, such as Word2vec, can also be interpreted with some simple vector arithmetics (e.g. element-wise addition), but we suppose that the property of word vectors mentioned above could be an even more straightforward interpretation regarding how information has been encoded in word vectors. This type of “interpretability” has been previously discussed in [1, 2].
>
> - FOFE model: Thanks for pointing out this inspiring reference. The idea of employing a constant forgetting factor to model word-order information is very interesting. In this regard, we implemented the FOFE model and tested it on both Yahoo and Yelp Polarity datasets. We experimented with different choice of the constant forgetting factor (\alpha):
>
> \alpha                 0.9        0.99      0.999     0.9999       1.0       SWEM-aver   SWEM-concat
> Yelp P.              84.58     93.01      93.81     93.79       93.48          93.59               93.76
> Yahoo! Ans      72.66     72.72     73.03      72.82       72.97         73.14                73.53
>
> It is worth noting that when \alpha = 1, FOFE is very similar to SWEM-aver model, except the fact that FOFE takes the sum over all words, rather than average. As shown above, with a careful selection of \alpha, FOFE can get slightly better performance on Yelp dataset (with \alpha = 0.999), compared to SWEM-concat. While on Yahoo dataset, we do not observe significant performance gains with the FOFE model. These results are in consistent with our observations that word-order features are necessary for sentiment analysis, but not for topic prediction. We will include this reference and the additional results in the revised version.
>
> Besides, we totally agree that developing sentence embeddings that are both simple and efficient is a very promising research direction (FOFE is a great work along this line).
>
> - Thanks for pointing out the wording and format issue. We will fix them accordingly in the revision.
>
> Hopefully our clarifications could address the concerns and questions raised in your review. Thanks!
>
> [1] Lipton, Zachary C. "The mythos of model interpretability." arXiv preprint arXiv:1606.03490 (2016).
> [2] Subramanian, Anant, et al. "SPINE: SParse Interpretable Neural Embeddings." arXiv preprint arXiv:1711.08792 (2017).

---

> ### Author Response · Authors · 2018-01-15
> **Additional results on sequence tagging tasks**
>
> Thanks for your update and valuable suggestion! We totally agree that sequence tagging should be a very important NLP problem to be considered, which could make the systematic comparisons in our paper more diverse and comprehensive. In this regard, we have tried on two (structured) sequence tagging tasks (i.e. chunking, NER). Specifically, we have considered the standard CoNLL2000 chunking and CoNLL2003 NER datasets. The corresponding results (F1 score) are shown as below:
>
>     Dataset              CNN-CRF [1]         BI-LSTM-CRF [2]        SWEM-CRF
>
> CoNLL2000                  94.32                      94.46                         90.34
>
> CoNLL2003                  89.59                      90.10                         86.28
>
> SWEM-CRF indicates that CRF is directly operated on top of the word embedding layer and make predictions for each word (there is no contextual/word-order information before CRF layer, compared to CNN-CRF or BI-LSTM-CRF). As shown above, CNN-CRF and BI-LSTM-CRF consistently outperform SWEM-CRF on both sequence tagging tasks, although the training takes around 4 to 5 times longer (for BI-LSTM-CRF) than SWEM-CRF. This suggests that for chunking and NER, compositional functions such as LSTM or CNN are very necessary, because of the sequential (order-sensitive) nature of sequence tagging tasks.
>
> One interesting future direction is to design some models that are simple yet still effective at capturing the contextual information needed for sequence tagging tasks. [3] is a great work along this line, which has proposed a simple and fast model for NER based on FOFE. We thank you again for pointing out the FOFE paper!
>
> All told: again, thanks for the helpful, critical feedback! We think that the paper, with these additional results, should have much more general implications in NLP than it was on submission, and sincerely hope you will agree.
>
>
> [1] Collobert, Ronan, et al. "Natural language processing (almost) from scratch." Journal of Machine Learning Research 12.Aug (2011): 2493-2537.
> [2] Huang, Zhiheng, Wei Xu, and Kai Yu. "Bidirectional LSTM-CRF models for sequence tagging." arXiv preprint arXiv:1508.01991 (2015).
> [3] Xu, Mingbin, and Hui Jiang. "A FOFE-based Local Detection Approach for Named Entity Recognition and Mention Detection." ACL 2017.

---

> ### Author Response · Authors · 2018-01-17
> **Additional experiments added**
>
> More experiments have been conducted for the sequence tagging tasks: we shuffled all the words within each input sentence (along with the corresponding labels) for the training set and trained a BI-LSTM-CRF model on both datasets. For NER, the F1 score drops from 90.10 to 85.79; while for chunking, the F1 score drops from 94.46 to 90.68. This observation indicates that the word-order information within a sentence does play an important role in sequence tagging problems, which is in consistent with our SWEM-CRF model’s results.
>
> With these additional investigations regarding the concerns you pointed out, we suppose that our contributions in general should now be much more solid. Looking forward to your feedback regarding our update, and we would be very much interested in an open discussion to find out if there are any remaining unfavorable factors. Thanks a lot for your time!

---

### Official Review · AnonReviewer4 · 2017-11-27
**A strong strong-baseline proposal**

**Rating:** 7
**Confidence:** 4

**Review:**

This paper presents a very thorough empirical exploration of the qualities and limitations of very simple word-embedding based models. Average and/or max pooling over word embeddings (which are initialized from pretrained embeddings) is used to obtain a fixed-length representation for natural language sequences, which is then fed through a single layer MLP classifier. In many of the 9 evaluation tasks, this approach is found to match or outperform single-layer CNNs or RNNs.

The varied findings are very clearly presented and helpfully summarized, and for each task setting the authors perform an insightful analysis.

My only criticism would be the fact that the study is limited to English, even though the conclusions are explicitly scoped in light of this. Moreover, I wonder how well the findings would hold in a setting with a more severe OOV problem than is perhaps present in the studied datasets.

Besides concluding from the presented results that these SWEMs should be considered a strong baseline in future work, one might also conclude that we need more challenging datasets!

Minor things:
- It wasn't entirely clear how the text matching tasks are encoded. Are the two sequences combined into a single sequence before applying the model, or something else? I might have missed this detail.

- Given the two ways of using the Glove embeddings for initialization (direct update vs mapping them with an MLP into the task space), it would be helpful to know which one ended up being used (i.e. optimal) in each setting.

- Something went wrong with the font size for the remainder of the text near Figure 1.

** Update **
Thanks for addressing my questions in the author response.

After following the other discussion thread about the novelty claims, I believe I didn't weigh that aspect strongly enough in my original rating, so I'm revising it. I remain of the opinion that this paper offers a useful systematic comparison that goes sufficiently beyond the focus of the two related papers mentioned in that thread (fasttext and Parikh's).

---

> ### Author Response · Authors · 2017-12-24
> **Authors' response to review 4**
>
> Thanks for your positive review!
>
> - For the text matching tasks, we first use a certain (single) compositional function (mean/max pooling, LSTM or CNN) to encode both sequences into two fixed-length vectors. Then we compare the two vectors by taking their concatenation, element-wise subtraction and element-wise product. These three features are concatenated together and further sent to an MLP classifier for prediction.
>
> - We found that the following tasks performed stronger empirically by mapping with an MLP, while keeping the Glove embeddings fixed: SNLI, MultiNLI, MR, SST-1, SST-2 and TREC. This will be included in future edition.
>
> - We agree that extending our investigation to other language would be an interesting future direction to pursue. Besides, we definitely need more challenging datasets (where higher-level semantic features can be leveraged) for a deeper understanding of natural language!
>
> - OOV problem: empirically, we found that the performance of SWEMs is not sensitive to the choice of vocabulary size, in other words, the number of OOV words. As discussed in the Supplementary, the key words used for predictions are typically of a frequency of around 200 to 300 in the training set. Therefore, we conjecture that treating those relatively rarely words (e.g. appear less than 50 times) as OOV would not have a big impact on the final results.
>
> - Thanks for pointing out. We will fix the font size issue in the revised version.

---

### Official Review · AnonReviewer3 · 2017-11-28

**Rating:** 6
**Confidence:** 5

**Review:**

This paper extensively compares simple word embedding based models (SWEMs) to RNN/CNN based-models on a suite of NLP tasks.
Experiments on document classification, sentence classification, and natural language sequence matching show that SWEMs perform competitively or even better in the majority of cases.
The authors also propose to use max pooling to complement average pooling for combining information from word embeddings in a SWEM model to improve interpretability.

While there is not much contribution in terms of technical novelty, I think this is an interesting paper that sheds new lights on limitations of existing methods for learning sentence and document representations.
The paper is well written and the experiments are quite convincing.
- An interesting finding is that word embeddings are better for longer documents, whereas RNN/CNN models are better for shorter text. Do the authors have any sense on whether this is because of the difficulty in training an RNN/CNN model for long documents or whether compositions are not necessary since there are multiple predictive independent cues in a long text?
- It would be useful to include a linear classification model that takes the word embeddings as an input in the comparison (SWEM-learned).
- How crucial is it to retrain the word embeddings on the task of interest (from GloVe initialization) to obtain good performance?

---

> ### Author Response · Authors · 2017-12-24
> **Authors' response to review 3**
>
> Thanks for your positive feedback!
>
> - According to our experiments, we tend to think that compositions are not as necessary for longer documents as for short sentences, which is the main reason that SWEM performs comparable or even better than RNN or CNN. The evidence here is two-fold: first, for Yelp review datasets, the text sequences considered are also long documents. However, since word-order features are necessary for sentiment analysis tasks (as demonstrated from multiple perspectives in the paper), CNN or LSTM has shown better results than SWEM. This indicates that even in the case of modeling longer text, LSTM and CNN could potentially take advantage of compositional (word-order) features if necessary. Second, we did observe that there are typically multiple key words (i.e. predictive independent cues) in a longer text for prediction, especially in the case of topic predictions (where the key words could be very topic-specific). This may intuitively explain why compositions are not necessary for document categorization.
>
> - Thanks for suggesting this! We agree that including a linear classifier comparison would be useful. In this regard, we trained and tested our SWEM-concat model along with a linear classifier (denoted as SWEM-linear), the results are shown as below (word embeddings are initialized from GloVe and directly updated during training):
>
> Model                           Yahoo! Ans.              Yelp P.
> SWEM-concat	           73.53                      93.76
> SWEM-linear                    73.18                      93.66
>
> As shown above, employing a linear classifier only leads to a very small performance drop for both Yahoo and Yelp datasets. This observation highlights that SWEM model is able to extract robust and informative sentence representations.
>
> - It is quite necessary to fine-tune the GloVe embeddings. As discussed in the paper, an intrinsic difference between GloVe and SWEM-learned embeddings is that the latter are very sparse. This is closely related to the fact that SWEM is utilizing the key words to make predictions. As a result, we would need to update GloVe embeddings or transform them to another space to boost the model performance. At the same time, we also found that for large-scale datasets (such as Yahoo or Yelp dataset), initializing with GloVe does not contribute a lot to the final results (i.e. randomly initializing the word embeddings leads to similar performance).

---

### Author Response · Authors · 2017-12-26
**Additional clarification regarding the novelty concern**

As for the novelty concern raised by the reviewer 1, we want to further highlight our contributions more clearly.

There are some recent works finding that CNN/LSTM may not be necessary for certain NLP problems. However, the general trade-offs among different compositional functions (simple operations versus more complicated ones) for various NLP applications have not been widely recognized yet and are far from systematic. Our work aims to bridge this gap by conducting an extensive comparative study on a wide range of text-representation-based tasks.

In this regard, the main contribution of our paper is not to achieve state-of-the-art results, but to investigate the relative importance of word embeddings and compositional functions, as well as to understand the observed results by unveiling the underlying reasons. Therefore, we keep the models to be compared as simple as possible, so that the functionality of different compositional functions could be highlighted.

Moreover, although max-pooling operation has been employed a lot along with convolutions in NLP, our utilization of max-pooling here is different in two main aspects: 1) as far as we are concerned, we are the first to apply max-pooling directly over word embeddings matrix; 2) this operation is shown to endow our SWEM-max model with improved transparency/interpretability (which is one major motivation of our work), and to extract complementary features with averaging operation as well.

To conclude, our work discovers several general rules (along with careful analysis) on how to rationally choose compositional functions for different NLP problems, which may let us rethink the necessity of employing CNN/LSTM in certain application scenarios. Besides, another interesting research direction, based on our findings, is to develop more challenging NLP datasets that require higher-level language understanding capabilities.

---

> ### Comment · Area_Chair · 2018-01-04
> **Novelty Claim**
>
> I want to push back on the novelty claims here. I think we are in agreement that 1) CNN with max pooling is widely used and shown to be effective, and 2) has been shown in many papers to yield greater interpretability. The claim here is that max-over time pooling with embeddings makes this novel. This feels like a stretch. At heart, embeddings are just a kernel-1 convolution. And BoW is just sum-over time pooling. While I don't have a reference for the exact use case of kernel-1 convolution with max-over-time pooling, it has very likely been tried before.

---

> > ### Author Response · Authors · 2018-01-07
> > **Authors' response**
> >
> > As stated above, the main contribution of our paper is to discuss the general trade-offs among distinct compositional functions for various NLP tasks. Besides, we propose, for the first time, to apply max-pooling operation (as a new type of compositional function) directly over the word embedding matrix and have demonstrated its advantages (performance gains, interpretability). To the best of our knowledge, the use of max-pooling operation alone as a compositional function has not been explored before. If possible, could you please let us know the reference that has tried the same setup as our SWEM-max model?

---

> > > ### Comment · Area_Chair · 2018-01-07
> > > **Confusion**
> > >
> > > My claim is that this is a semantic distinction. Why wouldn't a kernel size-1 convolution of the same # of features as embedding size, perform as roughly as fast (asymptotically), have roughly the same number of parameters, and perform at least as well your methods? If the kernel was an identity, wouldn't this CNN be exactly the same as SWEM. And of course max-pooling, min-pooling, sum-pooling have all been tried extensively in the single layer CNN context.

---

> > > > ### Author Response · Authors · 2018-01-08
> > > > **Re: Confusion**
> > > >
> > > > We argue that good papers are not always about designing novel algorithms. In the extreme case, you can think of SWEM as a special case of CNN. You can even think of SWEM-aver as a special case of RNN where the transition function is just an adding operation. However, there is no doubt that SWEM models are much simpler than CNN or LSTM, in terms of both computational complexity/speed and number of parameters, but they typically ignore the sequential/compositional information (e.g. word-order features). From this perspective, there is not much work that has investigated the trade-offs among different compositional functions. Our work aims to understand this important research problem with solid experiments and careful analysis.
> > > >
> > > > Thus, the motivation of our paper is not to claim that we develop a new model/algorithm, but to discuss/understand the general trade-offs stated above and to answer the following research questions: when is it necessary to employ more complicated compositional function, such as LSTM or CNN, for various NLP tasks? What information, other than the semantic meaning of individual words, is needed for distinct problems? Given the observation that SWEM performs very strong on text matching and document categorization, what semantic features are taken advantage of by SWEM to make the final predictions? How robust are different compositional functions with a relatively small number of training data?  We did not know the answer to these questions before our investigation. Max-pooling is introduced while we are trying to answer the third question, and it turns out to help us understand how SWEM works and boost the SWEM performance as a side benefit.

---

> > > > > ### Comment · Area_Chair · 2018-01-08
> > > > > **Re: Confusion**
> > > > >
> > > > > Sure, I'm all for simplicity on text matching. But your claim goes further, you say: "Surprisingly, SWEM consistently outperforms CNN and LSTM models by a large margin, on a wide range of training data proportions. "
> > > > >
> > > > > I get that your hyperparameters may be better than past experiments. (And Parikh has shown that simple word-based models do really well on these problems).
> > > > >
> > > > > But what is going on here?  We are in agreement that a single-layer-thin CNN should certainly be able to replicate the SWEM results. (And if you count embedding parameters (which I think you should), has roughly the same number of parameters.) So why is it scoring so much worse in these experiments? It shouldn't hurt so much to have CNN/ngrams vs SWEM/unigrams.

---

> > > > > > ### Author Response · Authors · 2018-01-09
> > > > > > **Re: Confusion**
> > > > > >
> > > > > > As to the claim that ‘SWEM consistently outperforms…, on a wide range of training data proportions’, we are considering the case where only part of the training data is available. For example, as shown in Figure 2, for both Yahoo! Ans and SNLI datasets, SWEM consistently performs much better than CNN or LSTM on the wide range of 0.1% ~ 10% proportion of original training data. With the whole training set, SWEM typically performs comparable or a bit better than LSTM or CNN on text matching and document topic prediction tasks. This indicates that SWEM is much less likely to overfit with limited training observations.
> > > > > >
> > > > > > For LSTM and CNN, most of our results are directly used from previous literature wherever available. As to SWEM, there are not much hyperparameters to be tuned thanks to its simplicity and our reported results are quite robust to the selection of hyperparameters. We will make our code publicly available after publication.
> > > > > >
> > > > > > The reasons that SWEM outperforms LSTM or CNN in some cases could be two-fold: 1) as already discussed in the paper, because of the simplicity, SWEM could be much easier to be optimized and thus may converge to some better local optima; 2) as suggested in [1], simpler methods tend to be better at capturing semantics than RNN’s and LSTM’s, although ignoring word-order information. Therefore, for datasets where word-order information is not important (such as Yahoo! Ans or SNLI), directly optimizing the word embeddings (semantics), as in SWEM, could be a better strategy.
> > > > > >
> > > > > > [1] Arora, Sanjeev, Yingyu Liang, and Tengyu Ma. "A simple but tough-to-beat baseline for sentence embeddings." (2016).

---

> > > > > > > ### Comment · Area_Chair · 2018-01-09
> > > > > > > **Understood**
> > > > > > >
> > > > > > > Okay, I understand your claim.
> > > > > > >
> > > > > > >
> > > > > > > - While people have unfortunately made exaggerated claims about LSTMs or 29-layer CNNs for these tasks, I think most people in NLP use word-based models or very thin one layer CNNs. My worry is that you are emphasizing the use of big models, but ignoring an influential set of results on the same tasks with single word or bigram models, e.g. like fasttext (https://arxiv.org/pdf/1607.01759.pdf) or Parikh's results for SNLI.
> > > > > > >
> > > > > > > - Can you make it clear which CNNs you are using? I agree with your conclusion that they are overfitting, but I want to be sure you are trying very simple CNNs for these tasks. For instance in fasttext (https://arxiv.org/pdf/1607.01759.pdf) with a bigram (kernel 2 model) they get similar really good results on the document classification tasks.

---

> > > > > > > > ### Author Response · Authors · 2018-01-10
> > > > > > > > **Re: Understood**
> > > > > > > >
> > > > > > > > We agree and are well aware that most people are using very thin (one-layer) CNNs, rather than 29-layer CNNs, for NLP problems. We specifically mentioned in the introduction part that most of our comparisons were considering one-layer recurrent/convolutional models (except for document classification tasks where deep models’ results were available). Besides, although fasttext and Parikh’s results have manifested the advantages of simpler model on certain tasks, there were hundreds of recent papers on text representation learning that were based on LSTM or CNN compositional functions, without comparisons to simpler methods. In this regard, the general trade-offs among different compositional functions have not been widely recognized yet. There is a clear gap here for research.
> > > > > > > >
> > > > > > > > More importantly, the motivations of the two papers you mentioned are different from ours. As a result, we have presented a much more comprehensive investigation regarding the necessity of employing complicated compositional functions (LSTM or CNN) and have answered many research questions they did not discuss: when (on what type of tasks) do simpler methods work better? When are CNN or LSTM-based models necessary?  Why are the advantages provided by complicated compositional functions so limited on tasks such as text matching or document categorization, in other words, why are simpler methods so efficient on these problems? Neither the fasttext paper nor Parikh’s work has explored these interesting questions.
> > > > > > > >
> > > > > > > > Besides, from Parikh’s results, we cannot directly draw the conclusion that simplicity is better, because the superior results they got may stem from the fact that the compare-aggregate framework they proposed is very efficient, which has made LSTM or CNN unnecessary. Moreover, they have only shown results on SNLI dataset, so that their observations may not apply to other text matching problems in general (e.g. paraphrase identification, answer sentence selection).
> > > > > > > >
> > > > > > > > Moreover, fasttext and our SWEM variants all belong to the category of simpler methods (with parameter-free compositional functions). Since our motivation is to explore the necessity of employing complicated compositional functions for various NLP tasks, we do not think it is necessary for us to make any comparisons between fasttext and SWEM.

---

### Decision · Program_Chairs · 2018-01-29
**ICLR 2018 Conference Acceptance Decision**

**Decision:**

Reject

**Comment:**

This work presents a strong baseline model for several NLP-ish tasks such as document classification, sentence classification, representation learning based NLI, and text matching. In terms of originality, reviewers found that "there is not much contribution in terms of technical novelty" but that "one might also conclude that we need more challenging dataset". There was significant discussion about whether it "sheds new lights on limitations of existing methods" or whether the results were "marginally surprising". In terms of quality, reviewers found it to be an "insightful analysis" and noted that these "SWEMs should be considered a strong baseline in future work".

There was significant discussion with the AC about the signficance of the work. In the opinion of the AC reviewers did were too quick to accept the authors novelty claims, and did not push them enough to include other baselines in their tables that were not overly deep model. In particular the AC felt that important numbers were left out of the experiment tables, for document classification that muddied the results. The response of the authors was:

"Moreover, fasttext and our SWEM variants all belong to the category of simpler methods (with parameter-free compositional functions). Since our motivation is to explore the necessity of employing complicated compositional functions for various NLP tasks, we do not think it is necessary for us to make any comparisons between fasttext and SWEM."

In addition when a reviewer pointed out the lack of inclusion of FOFE embeddings, the authors noted something similar

"Besides, we totally agree that developing sentence embeddings that are both simple and efficient is a very promising research direction (FOFE is a great work along this line)."

The reviewer correctly pointed out related work that shows a model very similar to what the author's propose. In general this seems like evidence that the techniques are known, not that they are significant and novel.